

# Advancing retinoblastoma detection based on binary arithmetic optimization and integrated features

Nuha Alruwais[1], Marwa Obayya[2], Fuad Al-Mutiri[3], Mohammed Assiri[4], Amani A. Alneil[4] and Abdullah Mohamed[5]

[1] Department of Computer Science and Engineering, College of Applied Studies and Community Services, King Saud University, Saudi Arabia, Riyadh, Saudi Arabia
[2] Department of Biomedical Engineering, College of Engineering, Princess Nourah bint Abdulrahman University, Riyadh, Saudi Arabia
[3] Department of Mathematics, King Khalid University, Muhayil Asir, Saudi Arabia
[4] Department of Computer Science, Prince Sattam Bin Abdulaziz University, Aflaj, Saudi Arabia
[5] Research Centre, Future University, New Cairo, Egypt

Corresponding author
Mohammed Assiri,
m.assiri@psau.edu.sa

## ABSTRACT

Retinoblastoma, the most prevalent pediatric intraocular malignancy, can cause vision loss in children and adults worldwide. Adults may develop uveal melanoma. It is a hazardous tumor that can expand swiftly and destroy the eye and surrounding tissue. Thus, early retinoblastoma screening in children is essential. This work isolated retinal tumor cells, which is its main contribution. Tumors were also staged and subtyped. The methods let ophthalmologists discover and forecast retinoblastoma malignancy early. The approach may prevent blindness in infants and adults. Experts in ophthalmology now have more tools because of their disposal and the revolution in deep learning techniques. There are three stages to the suggested approach, and they are pre-processing, segmenting, and classification. The tumor is isolated and labeled on the base picture using various image processing techniques in this approach. Median filtering is initially used to smooth the pictures. The suggested method's unique selling point is the incorporation of fused features, which result from combining those produced using deep learning models (DL) such as EfficientNet and CNN with those obtained by more conventional handmade feature extraction methods. Feature selection (FS) is carried out to enhance the performance of the suggested system further. Here, we present BAOA-S and BAOA-V, two binary variations of the newly introduced Arithmetic Optimization Algorithm (AOA), to perform feature selection. The malignancy and the tumor cells are categorized once they have been segmented. The suggested optimization method enhances the algorithm's parameters, making it well-suited to multimodal pictures taken with varying illness configurations. The proposed system raises the methods' accuracy, sensitivity, and specificity to 100, 99, and 99 percent, respectively. The proposed method is the most effective option and a viable alternative to existing solutions in the market.

# INTRODUCTION

Retinoblastoma, the most prevalent pediatric intraocular malignancy, can cause vision loss in children and adults worldwide. Adults may develop uveal melanoma. It is a hazardous tumor that can expand swiftly and destroy the eye and surrounding tissue. Thus, early retinoblastoma screening in children is essential. This work isolated retinal tumor cells, which is its main contribution. Next, stage and subtype the tumor. The methods let ophthalmologists discover and forecast retinoblastoma malignancy early. The approach may prevent blindness in infants and adults. Experts in ophthalmology now have more tools because of their disposal and the revolution in deep learning techniques. There are three stages to the suggested approach, and they are pre-processing, segmenting, and classification. The tumor is isolated and labeled on the base picture using various image processing techniques in this approach. Median filtering is initially used to smooth the pictures. The suggested method's unique selling point is the incorporation of fused features, which result from combining those produced using deep learning models (DL) such as EfficientNet and CNN with those obtained by more conventional handmade feature extraction methods. Feature selection (FS) is carried out to enhance the performance of the suggested system further. Here, we present BAOA-S and BAOA-V, two binary variations of the newly introduced Arithmetic Optimization Algorithm (AOA), to perform feature selection. The malignancy and the tumor cells are categorized once they have been segmented. The suggested optimization method enhances the algorithm's parameters, making it well-suited to multimodal pictures taken with varying illness configurations. The proposed system raises the methods' accuracy, sensitivity, and specificity to 100, 99, and 99 percent, respectively. The proposed method is the most effective option and a viable alternative to existing solutions in the market.

Orbital ultrasound, computed tomography (CT) of the orbit and head, chest X-ray, and bone scan are the diagnostic tools available for this condition. A yellowish-white or white cancer lesion on the fundus is visible during a manual ophthalmoscopic examination and is commonly linked to a rise in vascularization. The authors of this paper propose utilizing a backpropagation neural network to detect retinoblastoma in the retinal fundus.

Research done in 2015–16 indicates that over 8,000 children worldwide receive diagnoses with retinoblastoma annually, with 20% of all cases occurring in India since the uncontrolled mortality rate for such diseases in India is so high (*Dimaras et al., 2015*). Fluorescein angiography, ultrasound, computed tomography, and magnetic resonance imaging can all aid in diagnosing the condition (*Maitray & Khetan, 2017*). Nevertheless, pseudo-retinoblastoma situations may cause diagnostic mistakes when these symptoms are present. Monitoring the treatment's effectiveness and pinpointing the specific damage region is essential for assessing the tumors' growth after chemo reduction. In light of this, we offer an approach that seeks a precise disease diagnosis after each chemo dose decreases and extracts the tumor's exact growth. This approach aids medical professionals and expedites therapy according to patient health status.

Numerous research from various disciplines have demonstrated that tumor segmentation may be accomplished with high levels of detail using machine learning and

image processing. However, machine learning is a powerful tool for developing data-driven models because it can generalize commonalities between disparate human perspectives. Machine learning and image analysis are highly effective in the tumor segmentation challenge, particularly for liver (*Bai et al., 2019*) and lung (*Uzelaltinbulat & Ugur, 2017*), ophthalmology malignancies, and emotion analysis. Resid networks are highly effective at labeling candid pixels in X-ray images of liver cancer. Image processing techniques like thresholding, which are not data-driven, performed exceptionally well for segmenting lung tumors.

Based on machine learning, deep learning is a widely recognized and powerful method used to construct and create massive models with a foundation of exact networks of artificial neurons. Numerous ophthalmological investigations in recent years have made use of deep learning models. Research on topics like diabetic retinopathy and optical coherence tomography (OCT) (*Ting et al., 2019*) has demonstrated that they are valuable models for the detection of retinal disorders. Research in this area often involves some recognition or segmentation problems. In the identification task, supervised learning is typically used to identify a medical condition based on numerical information or a visual depiction of the person's eye.

The success of a classifier relies heavily on the features, making feature extraction a crucial stage in any image processing pipeline. While HC and DLM feature extraction can provide helpful information for categorization, not all features are created equal. Classifier efficiency may be diminished unintentionally by including unnecessary or duplicate characteristics. Consequently, this necessitates feature selection (FS), eliminating characteristics that might negatively impact the model's predictive accuracy to reduce the resulting space's dimensionality further. This issue of feature selection may be recast as an optimization problem, using loss minimization on the training dataset as the objective function. The problem may be stated formally as follows: reduce the loss on the training dataset while simultaneously optimizing the testing accuracy. Scientists currently employ metaheuristic techniques in various contexts, obtaining optimum solutions to real-world problems that were previously intractable or time-consuming to analyze using traditional algorithms. Due to the algorithm's inherent unpredictability, they can escape locally optimum solutions. This inspires our usage of a metaheuristic algorithm for FS. This research aims to develop an advanced and highly effective system for the early screening and diagnosis of retinoblastoma, the most common pediatric intraocular malignancy. This condition can lead to vision loss in children and adults and significantly threaten ocular health. The ultimate goal is to enhance the retinoblastoma detection and classification system's accuracy, sensitivity, and specificity.

In this work, we use a widely suggested metaheuristic method called the Arithmetic Optimization method (AOA) for the problem of FS. Here, we present a binary version of AOA called the Binary Arithmetic Optimization Algorithm (BAOA), which can be used in binary optimization situations like the FS problem and uses AOA's strengths. BAOA-S and BAOA-V are the names given to the two binary forms studied in this study; the specific contributions of this work in the context of retinoblastoma screening and diagnosis are as follows:

1. To address the issue of feature selection, we provide a hybrid deep learning EfficientNet-B0 gated recurrent unit (EffNet-GRU) model that outperforms the state-of-the-art in terms of accuracy, precision, sensitivity, and specificity. These two variations are designated as BAOA-S and BAOA-V, respectively.

2. Regularization is accomplished by using Navier–Stokes, which aids in picture smoothing and noise reduction.

3. The Gray-Level Co-occurrence Matrix (GLCM) is employed for hand-crafted feature extraction.

4. To address the issue of feature selection, we provide a hybrid deep learning EfficientNet-B0 gated recurrent unit (EffNet-GRU) model that outperforms the state-of-the-art in terms of accuracy, precision, sensitivity, and specificity. These two variations are designated as BAOA-S and BAOA-V, respectively.

5. A distinctive characteristic of the suggested strategy is the inclusion of fused features. Combining deep learning model outputs with manual feature extraction yields these characteristics. This fusion improves retinoblastoma detection and classification.

The article will proceed as described below. The relevant literature is briefly summarized in 'Related Works'. The article's usage of DL is briefly explained in 'Methods and Materials'. The effectiveness of the suggested method is evaluated through the presentation and discussion of experimental findings in 'Result and discussion'. 'Ablation Study' discusses the results and future directions.

## RELATED WORKS

Retinoblastoma is a widespread childhood eye malignancy. A total of 2–3% of all childhood cancers are it. Retinoblastoma's primary symptoms include leukocoria and strabismus. Retinoblastoma usually begins with leukocoria, a hazy white pupil. Intense light may make the pupil seem silvery or yellow. Squinted eyes, a wide pupil, a murky iris, and misaligned eyes are further signs. Terrible eyesight. This study reviews retinoblastoma, its features, past findings, and studies. *Allam, Alfonse & Salem (2022)* compare retinoblastoma detection AI methods. This AI study classified ocular tumors. Most researchers classify ocular tumors using pre-processing to remove noise and classifier. Classifiers use AI, machine learning, and computer vision. Backpropagation neural networks and image processing outperforms AI methods. Deep learning helps ophthalmologists detect and diagnose retinoblastoma promptly, according to *Deva Durai et al. (2021)* LPDMF filter pre-processes pictures, and CNN segments them. CNN segments retinoblastoma tumors and ocular anatomy.

A U-Net convolutional neural network (CNN) separates the foreground tumor cells from the background retinoblastoma segments in pre-processed pictures. The network augments data to enhance training samples with fewer restrictions. Manual filtering is necessary to confirm the prediction model given by the seven-layer U-Net architecture, even though the maps of features it uses increase geometrically at each level of the process (*Deva Durai et al., 2021*). *Henning et al. (2014)* developed a CNN approach to identify leukocoria, one of the critical signs of retinoblastoma. Leukocoria can induce cataracts, vitreous hemorrhages, and retinoblastoma. It may indicate systemic and intraocular diseases. The light-sensing

layer in the rear of the eye absorbs most light entering the pupil in leukocoria. However, the pupil reflects some light. Red reflex refers to the mirrored reddish-orange color. Flash photography causes red eye. One or both pupils may be white, yellow, or pale in leukocoria. A typical three-layer CNN trains 832 eye photos from afflicted children and Flickr, resulting in minimal accuracy with a limited dataset.

In addition, *Gupta, Garg & Kumar (2015)* studied the iris picture by employing a wavelet transform following the pre-processing median filtering and image augmentation. Histogram equalization and thresholding also differentiate between eye cancer and healthy eyes. *Henning et al. (2014)* conducted a study in 2014 to identify leukocoria in many photographs shot with a mobile phone. Without subjecting the digital picture of the eye to any pre-processing (apart from rescaling), the detection was performed through the iris using a CNN. The results of this study are reliable since they are based only on the learning procedure. *Rivas-Perea et al. (2014)* researched detecting leukocoria by iris analysis, and their method relies on median filtering to eliminate background noise. After denoising, an equalization histogram was carried out utilizing wavelet-based filtering and the Hough transform (*Rivas-Perea et al., 2014*) to improve angle detection. Diagnosing gastrointestinal diseases *via* the iris has been the subject of prior research using the backpropagation technique in medical imaging (*Syahputra et al., 2018*).

An image processing method was suggested by *Rivas-Perea et al. (2014)* to identify the position and size of the tiny circle in an eye picture that contains the iris. In an implementation, low error rates and high sensitivity were accomplished by using median filters and 2-dimensional stationary wavelet transformations. *Sofka & Stewart (2006)* merged the matching filter and vessel border parts. To quantify the vessels at each pixel, a mapping of this vector is developed using a meticulous training process. Significant gains were seen subjectively and numerically, with benefits shown in vessel centreline extraction occurring more quickly and with less effort. Machine learning techniques examined bladder cancer retinoblastoma gene mutations. A total of 18 RB1 mutation patients and 54 non-mutants had CTU scans. They chose features *via* analyzing Pearson's correlations and extracting features from sequences using a wrapper using XGBoost, RF, and KNN-generated models (*Ince et al., 2022*).

This method is commonly employed in ophthalmology and optometry to acquire precise and objective measurements of eyelid characteristics (*Liu et al., 2021*). The procedure entails considering the sketch's overall context, including the relationships between objects and their surroundings, to create realistic and visually appealing images (*Liu, Xu & Chen, 2023*). It focuses on improving the representation of both images and text by considering their specific features (characteristics) and their interrelationships (context) (*Yang et al., 2022*; *Gao et al., 2022*). These images are utilized to diagnose and assess diabetic retinopathy, a vision-related complication of diabetes. The deep learning system interprets these images and provides clinical evaluations, making it a valuable tool in the medical field (*Jin et al., 2023*; *Ye et al., 2022*).

The Gleason scoring system is used to assess the severity of prostate cancer (*Ao et al., 2023*). Visualizing the labeled Zika virus enables scientists to gain insights into its infection mechanisms and spread, which is crucial for understanding and developing treatments for

Zika virus infections (*Zheng et al., 2022*; *Wang et al., 2023*). This study likely utilizes data from the China Family Panel Studies to analyze how one family member's health status may impact other family members' health (*Hu et al., 2021*; *Liu et al., 2023b*) The central idea revolves around using IoT devices or sensors to collect data about the current situation or context, triggering or coordinating various services (*Cheng et al., 2016*; *Zhuang et al., 2022*) The differential sparse aspect likely involves leveraging differences between images or data points to aid the reconstruction process (*Tang et al., 2021*; *Zhang et al., 2022c*) yielding more precise and diagnostically valuable CT images while minimizing radiation exposure (*Lu et al., 2023b*; *Lu et al., 2023a*).

This modeling approach is crucial for simulating and comprehending heart tissue's intricate movements and deformations, with potential applications in medical research, diagnosis, and treatment planning, especially in cardiology and cardiac surgery (*Liu et al., 2023a*; *Guo et al., 2021*). The adhesive is formed by functionally (*Zhang et al., 2022a*). These biomarkers could prove invaluable in evaluating and monitoring the health effects of cadmium exposure in individuals (*Zeng et al., 2022*; *Wang et al., 2021*). CNNs have revolutionized computer vision tasks, achieving remarkable accuracy in image classification, face recognition, and object detection (*Li et al., 2021*; *Zhang et al., 2022b*).

CNN-based retinoblastoma detection approach employing computational image segmentation and convex polygon and area to determine tumor regression with 87% accuracy (*RL, Bauskar & Brahmapurkar, 2020*). While several researchers have developed automatic techniques for detecting osteosarcoma, the literature on these systems is scant compared to other cancer detection systems. Segmentation strategies for osteosarcoma in MRI and CT images range widely (*Altameem, 2020*; *Anisuzzaman et al., 2021*; *Nasor & Obaid, 2021*; *Zhang et al., 2018*). Using many different image processing methods, *Nasor & Obaid (2021)* suggested a method for segmenting osteosarcoma in MRI images. *Altameem (2020)* used intuitive fuzzy rank correlation-based segmentation to identify bone tumors in X-rays. After the data has been segmented, various statistical characteristics may be collected and analyzed using a deep neural network's layers and the Levenberg–Marquardt learning method.

## METHODS AND MATERIALS

This segment introduces a new hybrid deep learning model (EffNet-GRU) for detecting metastatic (tumor) cancer, along with its outline, architecture, and an assessment of its effectiveness in classification. In addition, an abstract depiction of the total classification system used in this work can be shown in Fig. 1. This classification method comprises data collection, pre-processing, model development, and assessment for identifying retinoblastoma.

The presented method for data augmentation is based on the targeted use of texture kernels. Texture kernels are small patches of image data used to generate new images by modifying the existing ones. This technique is used to increase the size of the training dataset and thus improve the model's performance. The latest data generated through this method can be used to train deep learning models and other machine learning algorithms.

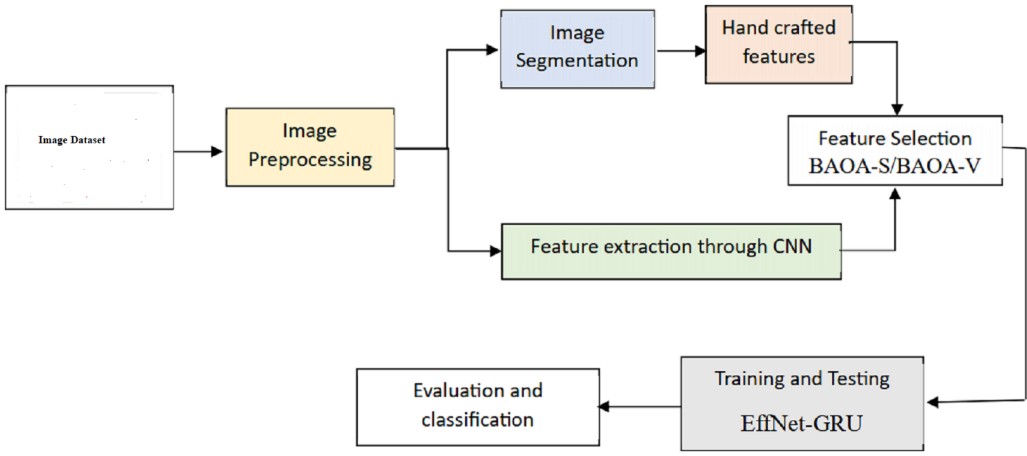

**Figure 1** Overall framework of the proposed model.

Eliminating eye-light spots in images is a common problem in computer vision. The Navier–Stokes equation, which describes the motion of fluids, can be used to build a technique for eliminating these spots. Navier–Stokes serves as a form of regularization, as it helps to smooth out the image and reduce noise. This technique can be combined with other methods to improve the overall performance of computer vision models. The concept behind this is to think of the picture as a fluid and then to let this fluid flow to fill the gaps, which are the areas specified on the mask. The morphological operation dilation is first performed to prepare for applying the Navier–Stokes inpainting. This action aims to guarantee that the holes and the black portion of the pupil are directly linked to one another. Consequently, the black pixels can move freely and fill the spaces.

The combined features approach involves generating features using both hand-crafted and deep-learning techniques. Hand-crafted features are those that experts manually design, while deep learning features are generated using neural networks. Combining these features and selecting the best among them can improve the performance of computer vision models. The issue of feature selection is a crucial aspect of computer vision models. The hybrid deep learning EfficientNet-B0 gated recurrent unit (EffNet-GRU) model provides a solution to this problem by using a combination of deep learning and recurrent neural network techniques. The model outperforms state-of-the-art algorithms regarding accuracy, precision, sensitivity, and specificity. The variations of this model, BAOA-S and BAOA-V, effectively address the feature selection issue. The proposed EffNet-GRU model was compared to other machine learning and deep learning algorithms using the same dataset, such as CNN-GRU, CNN-LSTM, and state-of-the-art categorization methods. The results showed that the EffNet-GRU model outperformed these algorithms regarding accuracy, precision, sensitivity, and specificity. This demonstrates the effectiveness of the proposed model in addressing the challenges of computer vision tasks.

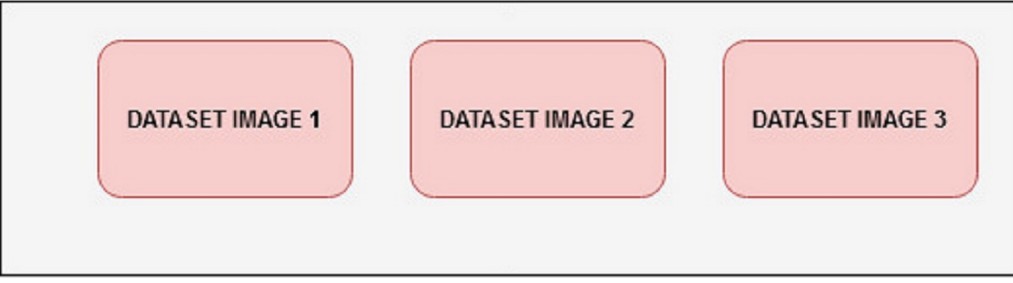

**Figure 2** Sample images from dataset 2.

## Dataset and pre-processing

We used images from the American Society of Retina Specialists' public dataset: http://imagebank.asrs.org. A dataset consisting of 2,048 photographs, 1,024 of which were healthy and 1,024 of which were ill, was acquired after the augmentation procedure. For testing, a dataset composed of 500 photos was utilized. In Dataset 1, the original dataset comprised 256 training samples, 500 test samples, and 500 validation samples, resulting in 1,256 samples. However, after applying data augmentation techniques, the training set was significantly expanded to include 1,024 samples, while the test and validation sets remained unchanged at 500 samples each, resulting in a new total of 2,048 samples. This augmentation process effectively quadrupled the size of the training dataset in Dataset 1.

Dataset 2 MRI images depict the scan's soft tissue delineation in the orbital and extraocular regions. Hyperreflective and hyperreflective retinoblastoma are shown in Fig. 2. The CT scan reveals the calcified area and nucleates the eye *via* the giant tumor. The fundus picture of the patients is captured using the Retcam pediatric camera. The planned study would focus on participants aged 12 to 20. Average diagnostic wait times range from zero to seven days. The experts are used to establish a reference point for assessing reality. A total of 243 pictures of pathogenic retinoblastoma are considered in the suggested technique shown in Fig. 1.

Moving onto Dataset 2, the initial dataset consisted of 1,920 training samples, 640 test samples, and 640 validation samples, totaling 3,200 samples. Post-augmentation, the training dataset experienced substantial growth, increasing to an actual 9,600 samples. Meanwhile, the test and validation sets remained unaltered, maintaining 640 samples each. Consequently, Dataset 2's total sample count swelled to 10,880 samples following data augmentation. This augmentation procedure greatly expanded the size of the training dataset within Dataset 2, facilitating enhanced model training and potentially improving overall model performance.

The presented method for data augmentation is based on the targeted use of texture kernels. Texture kernels are small patches of image data used to generate new images by modifying the existing ones. This technique is used to increase the size of the training dataset and thus improve the model's performance. The new data generated through this method can be used to train deep learning models and other machine learning algorithms. Eliminating eye-light spots in images is a common problem in computer vision. The

Navier–Stokes equation, which describes the motion of fluids, can be used to build a technique for eliminating these spots. Navier–Stokes serves as a form of regularization, as it helps to smooth out the image and reduce noise. This technique can be combined with other methods to improve the overall performance of computer vision models. The concept behind this is to think of the picture as a fluid and then to let this fluid flow to fill the gaps, which are the areas specified on the mask. The morphological operation dilation is first performed to prepare for applying the Navier–Stokes inpainting. This action aims to guarantee that the holes and the black portion of the pupil are directly linked to one another. Consequently, the black pixels can move freely and fill the spaces.

Accurate classification performance relies heavily on pre-processing. Data undergo this process before being sorted into categories. When training a model for classification on the "cancer data set", pre-processing techniques are crucial. Class classification, image scaling, data augmentation, random cropping, and sliding with the crop are some of the most used pre-processing methods. All photos were scaled to the exact dimensions to ensure the proposed model was trained with consistent results. The cancer data collection underwent data augmentation. Convolutional neural networks (CNNs) generally do better when dealing with large datasets, yet amassing such datasets can be difficult. Convolution neural networks have difficulty evaluating their performance because of an overfitting problem caused by insufficient training data. The best strategy for fixing the problem is to use a data augmentation strategy. The cancer dataset was also pre-processed using a technique called random cropping using convolution neural networks. For the model to be trained appropriately, a vast number of data must be made accessible throughout the process of randomly cropping various regions of high-dimensional pictures.

## Image segmentation

There is a great deal of data in the initial image, some irrelevant to the categorization process. They may also be considered background or irrelevant information since they may interfere with the classifier's forecast. Segmentation, a sort of pre-processing, is used to strip this data from the pictures. The final result of a segmentation process is an image from which irrelevant details have been deleted. Muscles, ink marks, folded/severed tissues, and hazy areas are all background details that need editing. The RGB picture is first converted to HSV to apply hue normalization, and then the hue channel is subjected to multi-level thresholding. After that, the picture is changed back into its original RGB format. K-means clustering segmentation for $K = 3$ (pink, blue, and white clusters) is applied to the RGB picture. While the pink and blue groups are typical and necessary for feature extraction, the white group is unnecessary and should be discarded. The white collection may be eliminated by selecting the neighboring pixels and coloring them white. The completed product of this process is a picture that has been edited by having its backdrop removed. When using hand-crafted feature extraction approaches, segmentation is crucial. Because DLs can dynamically segment to determine the relevant area in an image, classification is unnecessary when using DLs.

## Feature extraction

The Gray-Level Co-occurrence Matrix (GLCM) is widely employed in artificial intelligence and computational imaging for hand-crafted feature extraction. The occurrence of each pair of grayscale values in an image is represented by a matrix called the grayscale locus of the correlation matrix (GLCM). Its purpose is to record a picture's texture details. The spatial connections between adjacent picture pixels are analyzed to calculate GLCM. The GLCM tracks how often two pixels at a certain distance and orientation occur together. Parameterizing the GLCM calculation enables the management of both space and direction.

After the GLCM has been calculated, several texture characteristics may be derived. Image classification, segmentation, and object identification are just machine-learning tasks that may take advantage of these attributes as inputs.

The formula for calculating the GLCM is as follows:

Let $I$ be an image with $M \times N$, and let d be the offset distance. Let $G$ be the set of gray-level values in the image. The GLCM $C$ at offset $(dx, dy)$ is defined as:

$$C(i,j) = \{count(m,n) \ in \ I | I(m,n) \tag{1}$$

$$= i \ and \ I(m+dx, n+dy) \ = \ j\}, \ for \ i,j \ in \ G \tag{2}$$

where $count\{(m,n) \ in \ I | I(m,n) \ = \ i$ and $I(m+dx, n+dy) \ = \ j\}$ is the number of times that a pair of pixels with gray-level values $i$ and $j$ occurs at offset $(dx, dy)$ in the image.

Once the GLCM is calculated, various statistical measures such as contrast, correlation, energy, and homogeneity can be computed. These can be used as texture features for image classification and other tasks. Numerous fields have found success using GLCM-based texture characteristics, including medical imaging, remote sensing, and industrial inspection. However, it is essential to note that deep learning approaches that directly learn features from the data have essentially supplanted handmade features like GLCM-based ones. The CNN is used as the deep learning model to extract features automatically, as shown in Fig. 3.

We use a hybrid of the two techniques above for each image to jointly identify features, resulting in two sets of feature vectors. The first group comprises feature vectors derived from a combination of HC and CNN feature extraction. A 1,350-element feature vector was constructed. The second collection includes CNN feature and HC feature vectors fused. The resulting feature vector has a size of 2,118 bytes. As a result, each image has access to a massive set of features. However, not all these details factor into the ultimate categorization. Therefore, selecting meaningful characteristics from the extracted features is essential. Both the BAOA-S and BAOA-V are described here.

## Binary arithmetic optimization algorithm (BAOA-S/BAOA-V)

The Binary Arithmetic Optimization Algorithm (BAOA) was initially presented to solve optimization issues using binary variables. The BAOA algorithm comes in a standard and a variable form. The BAOA-S algorithm keeps track of potential answers in binary strings. After each cycle, the algorithm applies a set of genetic operators on the current population
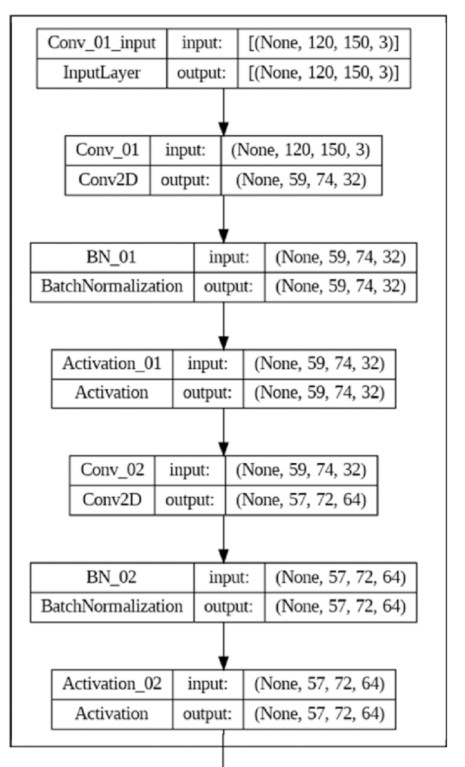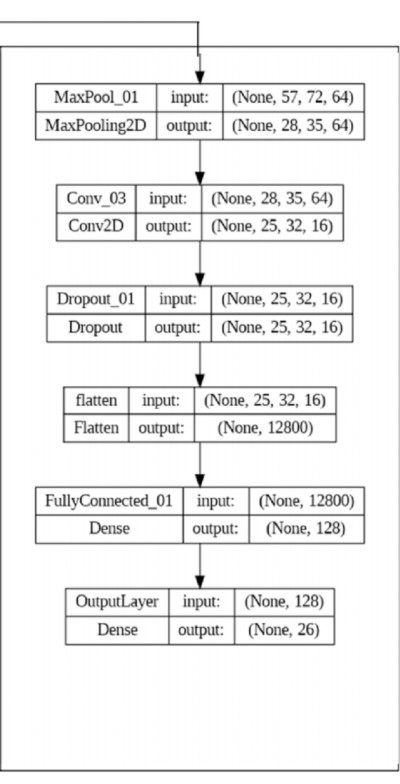

**Figure 3** The CNN model set up for feature extraction.

to develop a new set of possible solutions. The best solutions are then chosen to produce the next population once their fitness has been assessed. This procedure is continued until a stopping condition is reached, such as the maximum number of allowed iterations or a predetermined quality threshold for the result. The BAOA-V method expands the BAOA-S algorithm by supporting binary strings with varying lengths. This is helpful when solving optimization issues where the ideal number of choice variables may change from problem to problem. Optimization issues in fields as diverse as feature selection, machine learning, and engineering design have benefited from both iterations of the BAOA method. Comparisons to other metaheuristic algorithms, such as genetic and particle swarm optimization, have revealed that the approach performs competitively.

In a d-dimensional search space, starting solutions, denoted by $[x_m^1, x_m^2, \ldots, x_m^n]$, are created at random.

$$x_m^n = x_{min}^n + s(x_{max}^n - x_{min}^n) = \{1, 2, \ldots, M\}, n = \{1, 2, \ldots, n\} \tag{3}$$

In this formula, $M$ is the total number of individuals, $x_m$ is the index of the $m$th solution, $x_m^n$ is the index of the $n$th dimension of the $m$th solution, $x_{max}^n$ and $x_{min}^n$ are the maximum and minimum values of the jth dimension of the search space, and $s$ is a random number between 0 and 1. Matrix representations of the initial solution $Y$ are also possible, as

demonstrated by Eq. (4).

$$Y = \begin{bmatrix} x_1^1 & \cdots & x_1^n \\ . & \cdots & . \\ . & \cdots & . \\ . & \cdots & . \\ x_M^1 & \cdots & x_M^n \end{bmatrix} \tag{4}$$

Math Optimizer Accelerated (MOA) function is used to decide between exploration and exploitation.

$$M(L_i) = Min + L_i \times \left( \frac{Max - Min}{N_i} \right) \tag{5}$$

$L_i$ represents the current iterations, the maximum number of iteration is denoted by $N_i$. The lowest and highest values are denoted by $Min$ and $Max$. We use the dividing and multiplying operators to probe the solution space during this stage. Exploration is carried out by selecting, with equal probability, either the division or multiplication operator. A new answer is computed.

$$x_m^n(L_i + 1) = \left\{ Best(x^n) \% (M + \in) \times \left( (x_{max}^n - x_{min}^n) \times \mu + x_{min}^n \right), s_2 \right\} \tag{6}$$

In this exploitation stage, we are essentially doing a deep search for the ideal answer, or one that is very close to the best possible solution. As a result, operations such as addition and subtraction are utilized. The odds of getting picked as an exploitation operator are the same as during exploration. The new answers are found by,

$$x_m^n(L_i + 1) = \left\{ Best(x^n) - M \times \left( (x_{max}^n - x_{min}^n) \times \mu + x_{min}^n \right), s_3 \right\} \tag{7}$$

**Algorithm 1: Binary Arithmetic Optimization Algorithm (BAOA-S/BAOA-V)**

BAOA-S:

1. **Initialize** the binary string of the input number x and set the iteration count $i = 0$.
2. **Evaluate** the fitness function $f(x)$ andstore it as $f_{best} = f(x)$.
3. **Repeat** the following steps until the termination condition is met:
   (a) **Select** a random bit position b in $x$.
   (b) Flip the bit at position b to obtain a new $x_{new}$.
   (c) **Evaluate** the fitness function $f(x_{new})$.
   (d) **If** $f(x_{new}) > ff_{best}$, *set* $x = x_{new}$ *and* $f_{best} = f(x_{new})$.
   (e) **Increment** the iteration count $i$.
4. **Return** the optimized binary string x and its corresponding fitness value $f_{best}$.

BAOA-V:

1. **Initialize** the input number $x$ and set the iteration count $i = 0$.
2. **Evaluate** the fitness function $(x)$ andstore it as $f_{best} = f(x)$.
3. **Repeat** the following steps until the termination condition is met:
   (a) **Generate** a random vector v with the same length as x.
   (b) **Generate** a new binary $x_{new}$ by performing a bitwise XOR between x and $v$.
   (c) **Evaluate** the fitness function f(x_new).

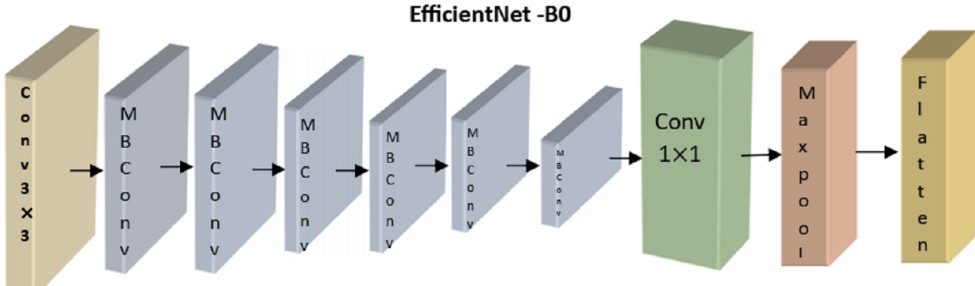

**Figure 4 Overview of EfficientNetB0's architecture.**

   (d) **If** $f(x_{new}) > f_{best}$ , *set* $x = x_{new}$ *and* $f_{best} = f(x_{new})$.

   (e) **Increment** the iteration count $i$.

  4. **Return** the optimized $x$ and its corresponding fitness value $f_{best}$.

The fitness function $f(x)$ depends on the specific problem being optimized and is not specified in the pseudocode. The termination condition may also be based on a maximum number of iterations, reaching a desired fitness threshold, or other criteria.

## Classification

After the completion of the pre-processing processes, the following step is to load the data into the model that has been proposed (EffNet-GRU). This section overviews the EffNet and GRU models and explains the learning parameters for cancer detection and classification.

The analysis module uses a recurrent and convolutional neural network to classify retinoblastoma. The CNN uses a lightweight, accurate model presented by the EfficientNetB0 architecture (*Deng et al., 2009*). EfficientNetB0 provides the smallest model in the EfficientNet family and can be processed more quickly than larger models. EfficientNetB0 is the best model to use when the model has to rapidly-produce a forecast since the differences in accuracy when upgrading to a higher model are small in our area. To improve the model's performance, we employ transfer learning (*Andayani et al., 2019*), in which we re-use weights that have proven successful in the past when applied to the dataset's object recognition task. Each of the nine blocks that comprise the EfficientNetB0 architecture comprises numerous layers. An overview of this architecture, including its nine significant components, is shown in Fig. 4.

Here, we concentrated on optimizing the correctness of the transferred model by adjusting the number of frozen layers. Four distinct configurations were used during training, each denoted by the number of layers constituting the EfficientNetB0 model. All layers are frozen and not trained from EfficientNetB0 in the no-training mode. Except for Block 9's pooling, flattening, dense, and dropout layers, all other layers are kept static throughout top training. Layers from the previous MBConv block, Block 8, are likewise trained in the partial version of the model (Block 9). All layers are kept unfrozen during the whole training.

The EffNet model relies heavily on the convolution layer. It is made up of a variety of trainable filters that can extract characteristics from raw data. The layer $k$ , is assumed to be identified as $x_b^c$, whereas the layer c-1, $m$th has been designated by $x_b^{c-1}$ in the feature map.

Following is the formula for determining $x_b^c$:

$$x_b^c = \sum_{m \in ij} a_{jm}^c * x_m^{c-1} + y_j^c \tag{8}$$

where $|i|$ is the total number of mapping features in layer $c$, $y_j^c$ is a biased term applied to all links leading to the $j$ feature map, and $ij$ is a collection of characteristic maps in layer $c-1$ that are associated with unit $j$. The expression for ReLU Functions activation $d(x)$ is as follows:

$$d(x) = max(o, x) \tag{9}$$

where the range is from 0 to x. Following is a mathematical expression for the max_pooling:

$$x_{b^{c+1}} m^{c+1}, s = \frac{max}{0 \le b \le H, 0 \le m < K} a_{b^{c+1}}^c * H + i, m^{c+1} * w + m \tag{10}$$

An initial evaluation was conducted, with each configuration training for 25 epochs before being tested. Based on our analysis of the experimental training accuracy, we determined that top training was the optimal setup. To prevent information loss from earlier layers during training, the model's weights are frozen at each layer except the last block of layers (which consists of pooling, flattening, dense, and dropout layers). Here, we merge a convolutional neural network (CNN) with a recurrent neural network (RNN) by building a GRU (gated recurrent unit) recurrent neural network on top of the EfficientNetB0 architecture. The GRU network is fed photos and tasked with determining which features disclose whether or not the eye is cancerous. The final classifier receives the GRU's output and uses dense and dropout layers to assign a value between 0 and 1 to the malignant picture.As previously noted, EfficientNetB0 is implemented in the GRU module of Fig. 4. EfficientNetB0 provides a foundation of information, and the remaining layers are taught to apply that knowledge to classify cancer. The completed network layout is seen in Fig. 5.

GRU is superior to LSTM due to its three main gates and internal cell state. The GRU has the data locked up securely. The reset gate $(g)$ presents the previously known information, whereas the update gate $(u)$ provides historical and future data. Using the reset gate, the current memory gate keeps and remembers crucial data from the preceding computer state. The input modulation gate allows for the introduction of nonlinearity while maintaining the input's zero-mean features. The mathematical formulation of the basic GRU of static and dynamically modified gates is,

$$g_t = \sigma \left( A_t . W_{ag} + L_{t-1} . W_{lg} + b_g \right) \tag{11}$$

$$y_t = \sigma \left( A_t . W_{ay} + L_{t-1} . W_{ly} + b_y \right) \tag{12}$$

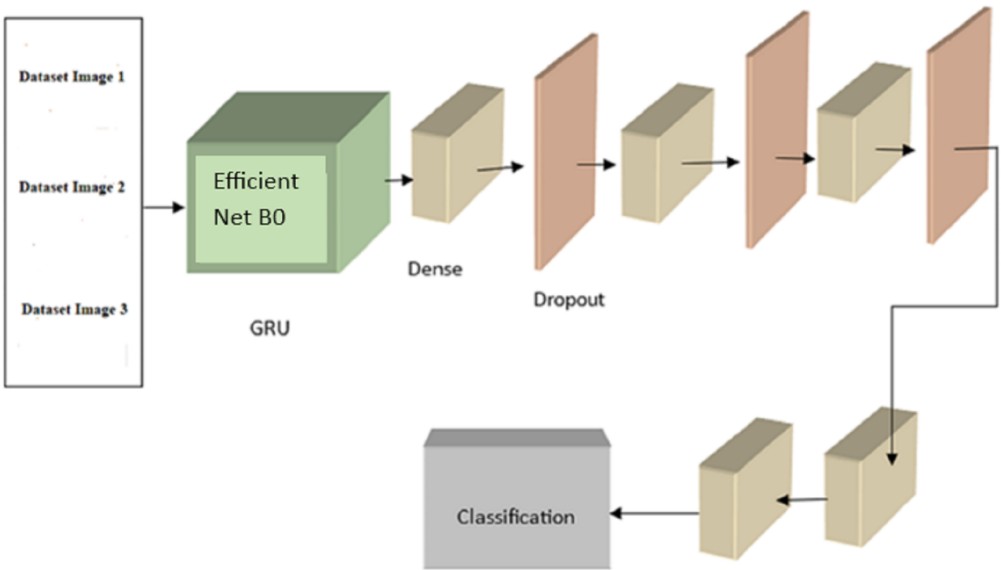

**Figure 5  The proposed EffNet-GRU model architecture.**

$W_{lg}$ and $W_{ly}$ are the weights, $b_g$ and $b_y$ are the bias.

The original picture input geometry was (48, 48, 3), having a height of 48 pixels and a width of 48 pixels in RGB and three channels. The suggested model uses a single convolutional layer to derive characteristics of the input shape, with a 64-shaped feature map as the layer's final output. In addition, the convolutional layer has a stride of $(3 \times 3)$ and a kernel size of 1. While typical layer one padding was maintained throughout the proposed model, variability was reduced by using rectified linear units (ReLU) as the activation function. The first convolutional layer's output was a shape with (48, 48) sized feature maps, totaling 64 features.

The suggested model's training process is sped up thanks to the pooling layer, which reduces the training parameter to the (58, 58) size. The training parameters (58, 58, 64) were sent *via* a dropout layer after the pooling layer to avoid overfitting. To prevent overfitting, the convolution layer was given an initial dropout of 0.5. The training parameter was drastically reduced after each conventional and max-pooling layer, and activation function (ReLU) and dropout were added. A fully connected layer built by flatten with training parameters (42, 42) size and features map (256) must have its input data combined into an ID array once the conventional and max-pooling layers have been trained. Table 1 shows the hyperparameter settings used in the proposed model. The dropout generated 512 feature maps once the convolutional layers procedure was finished. A GRU model was built with a fully linked layer of 512 neurons to address the vanishing gradient issue. The GRU paradigm was followed by the adoption of two ultimately linked layers. The last connected layer also includes SoftMax functions.

The most commonly used method for splitting data randomly is known as random splitting. In this study, the dataset was randomly shuffled to ensure unbiased partitioning.

**Table 1  Hyper parameter setting for the proposed model EffNet-GRU.**

| Model | Hyperparameter | Range of values to try | Finalised parameter |
|---|---|---|---|
| EfficientNet | input_shape | [height, width, channels] | (48,48,3) |
| | depth_multiplier | 0.5, 0.75, 1.0, 1.25, 1.5, 1.75, 2.0 | 1.5 |
| | width_multiplier | 0.5, 0.75, 1.0, 1.25, 1.5, 1.75, 2.0 | 1.0 |
| | dropout_rate | 0.0, 0.1, 0.2, 0.3, 0.4, 0.5 | 0.5 |
| | input_scaling | 0.33, 0.5, 0.67 | 0.5 |
| GRU | num_units | 16, 32, 64, 128, 256, 512, 1024 | 256,512 |
| | dropout_rate | 0.0, 0.1, 0.2, 0.3, 0.4, 0.5 | 0.5 |
| | recurrent_dropout_rate | 0.0, 0.1, 0.2, 0.3, 0.4, 0.5 | 0.5 |
| | Activation | 'tanh', 'relu', 'sigmoid', 'linear','softmax' | softmax |
| | return_sequences | 0,1 | 0,1 |

The dataset was then divided into the desired proportions, with 70% allocated for training, 15%, for validation, and the remaining 15% for testing. This approach was employed to maintain the integrity and representativeness of the dataset during the experimentation phase. Libraries such as sci-kit-learn in Python are commonly utilized for random splitting.

The selection of optimization algorithms, learning rate schedules, and convergence criteria is crucial in achieving practical training of machine learning models. The Adam algorithm is a widely used adaptive optimization technique that integrates components from both momentum and RMSprop algorithms. The learning rates are adapted for each parameter individually. The learning rate schedule is a crucial component in the training process as it governs the dynamic changes in the learning rate. The importance of model convergence and stability must be addressed in research. After conducting the ablation study, the learning rate was adjusted to 0.0001.

## RESULT AND DISCUSSION

This part discusses the classification efficiency (accuracy, precision, sensitivity, and specificity), along with the methodology and proposed model (EffNet-GRU) for an improved comparison study. To carry out this experiment, we used a CPU with a Core i6 architecture from Intel and a graphics processing unit (GPU) manufactured by NVIDIA. In addition, the suggested model has undergone training thanks to the combination of Keras and the Python 3.8 programming environment. Ten-fold cross-validation is used in this study to evaluate the performance of a predictive model. A resampling method helps assess how well a machine learning model will generalize to new, unseen data. In each iteration, the model is trained on the training set and evaluated on the test set. Evaluation metrics such as accuracy, F1 score, precision, recall, etc. are calculated to assess the model's performance on the current fold. In this cross-validation experiment, the mean accuracy across all folds is 100%. This indicates that the model performed exceptionally well and consistently across all subsets of the data, which is an ideal outcome in model evaluation.

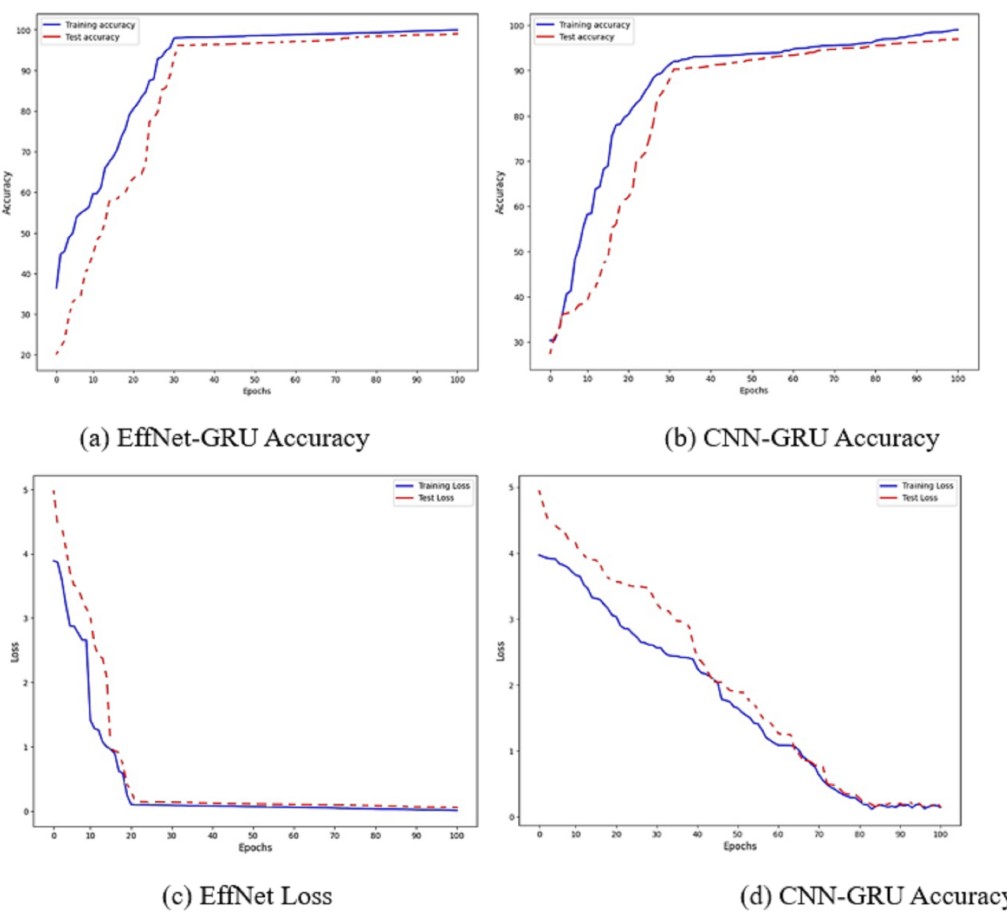

(a) EffNet-GRU Accuracy

(b) CNN-GRU Accuracy

(c) EffNet Loss

(d) CNN-GRU Accuracy

**Figure 6** **Accuracy and loss of the models EffNet-GRU and CNN-GRU.**

The expressions that follow are used to analyze the efficacy metrics:

$$Accuracy \ (A_c) = \frac{True_p + True_n}{True_p + True_n + False_p + False_n} \tag{13}$$

The accuracy of the proposed model is shown in Fig. 6. Figures 6A and 6B show the accuracy of the model EffNet-GRU and CNN-GRU. Figures 6C and 6D presents the loss of the model EffNet-GRU and CNN-GRU. Figure 6 shows that the efficient net with GRU performs better than other models.

$$Precision \ (P_r) = \frac{True_p}{True_p + False_p} \tag{14}$$

$$Recall \ (R_e) = \frac{True_p}{True_p + False_n} \tag{15}$$

$$F1 - Score \ (F_s) = 2 \times \frac{P_r * R_e}{P_r + R_e} \tag{16}$$

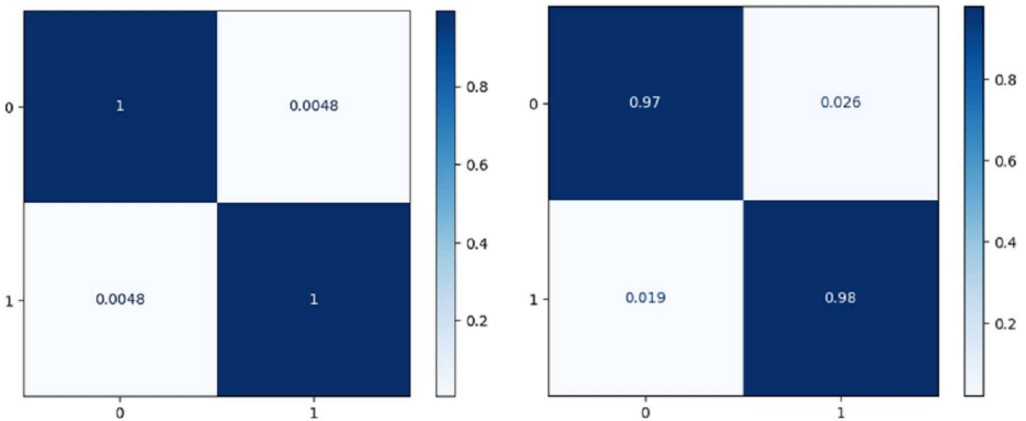

**Figure 7** Confusion matrix of the models EffNet-GRU and CNN-GRU.

**Table 2** Performance comparison of the different models on the same dataset.

| Model | Sensitivity | Specificity | Accuracy |
|---|---|---|---|
| SVM | 85.1% | 84.3% | 82.5% |
| CNN | 93.2% | 92.5% | 93.1% |
| EffNet | 96.2% | 96.8% | 97.2% |
| LSTM | 95.0% | 95.8% | 96.2% |
| CNN-GRU | 97.8% | 96.5% | 97.8% |
| EffNet-GRU | 99.8% | 99.7% | 100% |

Figure 7 shows the confusion matrix on the proposed model and CNN-GRU. The suggested method is sufficiently thorough to recognize a single malignant tumor exhibiting no variation. The algorithm is sensitive to discovering pixels that are not cancerous. The cluster of tumor pixels separately from other areas is also competent. According to the results of the suggested technique, which are presented in Table 2 and Fig. 8, the approach suggested has a sensitivity of 99.8%, a specificity of 99.7%, and an accuracy of 100% when analyzing both normal and abnormal clinical pictures.

The prior phase in establishing the parameters that are utilized in the propagation process is to perform research on the learning parameter rate chosen in the training process. These can be seen in Fig. 8, which is the prior phase in the procedure.

A study was carried out with the maximum value set at 100, and the learning period was given a range of varied matters. Because the outcomes of training using the learning rate of 0.0001 yielded an accuracy of 100% and an actual epoch that was considerably shorter than those obtained using any of the remaining learning rates, the learning rate of 0.0001 is now being utilized as a variable in the backpropagation method. It is demonstrated that the suggested algorithm outperforms previous state-of-the-art approaches by reliably matching photos in noisy, real-world settings while avoiding the difficulties introduced by the wide range of possible lighting situations. The result is a low-priced platform that can be employed in locations with sparse medical resources. The algorithm's accuracy can be

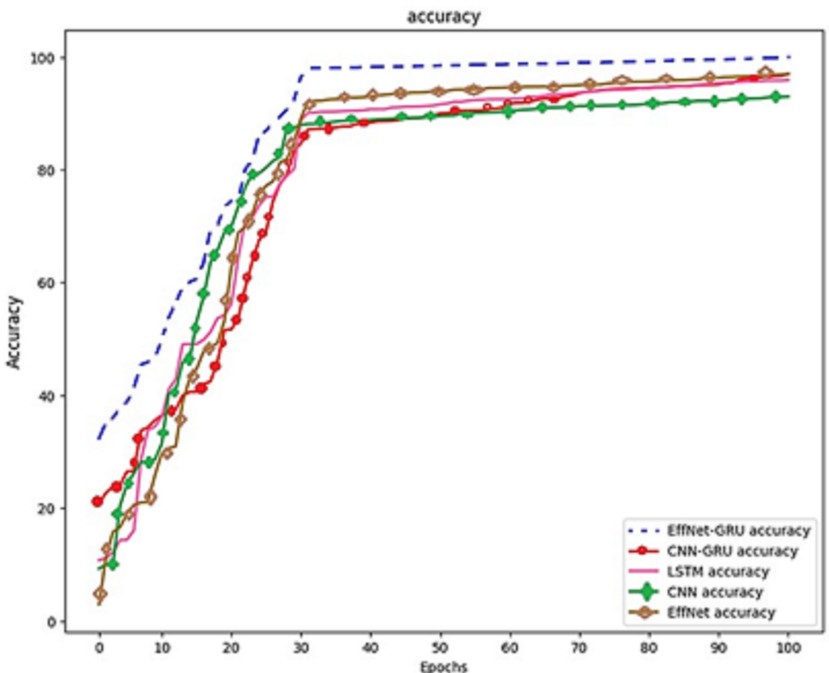

**Figure 8** **Model comparison on the same dataset.**

improved by developing a website where such algorithms are freely and quickly accessible to the general public to make accurate illness diagnoses.

Images of the fundus have been used as the foundation for creating artificial intelligence models of various retinal disorders, such as diabetic retinopathy, age-related macular degeneration, and retinopathy of prematurity. EfficientNet and GRU have demonstrated that they can detect normal and abnormal situations using fundus pictures with a sensitivity of 99.8% to 100% and a specificity of 99.7% to 99.8% in Fig. 9. This was accomplished by utilizing AI and ML to screen for retinoblastoma. Neural network-based analysis of retinal images is a relatively new field. Although studies have been conducted on removing retinal disorders, the peak of this technology is still yet to come. However, neural networks based on unsupervised learning are making some headway in processing retinal pictures. In addition, neural networks can have any number of layers and be designed in any way, with the network architecture being decided heuristically depending on the problem's domain. It is possible to employ many variants of deep neural networks to extract retinal anatomical features, including CNN, LSTM, EfficientNet, and GoogleNet.

In the quest to enhance a model's generalization capabilities and combat the menace of overfitting, several strategic practices come to the forefront. Among these, L1 and L2 regularization emerge as potent tools. By integrating L1 (Lasso) or L2 (Ridge) regularization terms into the loss function, the model's propensity to latch onto overly intricate patterns is curbed, effectively diminishing the risk of overfitting. Additionally, the judicious use of early stopping fortifies the model's resilience against overfitting. This technique entails monitoring the validation loss throughout training, promptly halting the process should

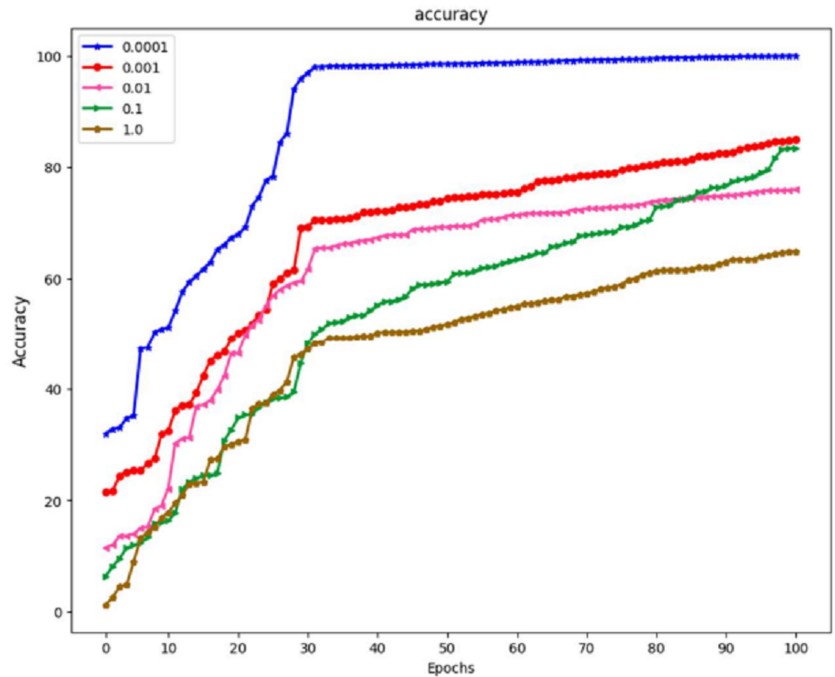

**Figure 9** **The effect of learning rate on the model EffNet-GRU.**

**Table 3** **Performance comparison of various methods used in the literature survey.**

| Reference | Methodology | Accuracy |
|---|---|---|
| *Andayani et al. (2019)* | Backpropagation neural network% | 90% |
| *Kaliki et al. (2023)* | Machine learning% | 98% |
| *Pratt et al. (2016)* | CNN% | 96.8% |
| *Worrall, Wilson & Brostow (2016)* | deep CNN% | 93.6% |
| *Deva Durai et al. (2021)* | CNN% | 99.8% |

it exhibit signs of degradation. A third pivotal strategy revolves around hyperparameter tuning. By subjecting variables such as the learning rate, batch size, and network architecture to systematic experimentation and fine-tuning, the model's capacity to generalize is profoundly influenced. Finally, to validate and ensure consistent performance, 10-fold cross-validation techniques are employed. This approach assesses the model's proficiency across various subsets of the dataset, thus validating its robustness and averting undue reliance on a single, random data split. Together, these practices form a formidable arsenal for safeguarding a model's capacity to generalize effectively while mitigating the peril of overfitting. Table 3 shows the performance comparison table with the existing methodology.

In a study by *Andayani et al. (2019)* a backpropagation neural network was employed to address a specific task or dataset. The researchers reported a noteworthy accuracy rate of 90% in their findings. In a study by *Kaliki et al. (2023)*, a high accuracy rate of

**Table 4** The performance result based on feature extraction techniques.

| Feature extraction techniques | Accuracy | F1 score | Precision | Recall |
|---|---|---|---|---|
| Automatic feature extraction | 88.5% | 0.875% | 0.890% | 0.862% |
| Handcrafted features | 86.2% | 0.861% | 0.873% | 0.849% |
| Fusion of feature extraction | 100% | 0.99% | 0.99% | 0.99% |

98% was achieved, suggesting a robust performance. In a survey by *Pratt et al. (2016)*, a convolutional neural network (CNN) was employed, yielding an accuracy rate of 75%. In recent years, convolutional neural networks (CNNs) have emerged as a popular choice for various image-related tasks. In a study conducted by *Worrall, Wilson & Brostow (2016)*, a deep convolutional neural network (CNN) was employed, suggesting the utilization of a neural network architecture with increased complexity. The findings showed that the observed accuracy rate was 93.6%. In a study by *Deva Durai et al. (2021)*, a convolutional neural network (CNN) was utilized to achieve a remarkable accuracy of 99.8%. This outcome suggests the model exhibited exceptional performance when applied to the task or dataset under investigation, performance comparison is provided in Table 3.

## ABLATION STUDY

### Based on based on feature extraction

In Automatic Feature Extraction from the data, the results show that this approach achieved an accuracy of 88.5%, which means it correctly classified 88.5% of the instances in the dataset.

Table 4 shows that the results indicate that the model using handcrafted features achieved an accuracy of 86.2%. It performed slightly lower than the automatic feature extraction method in terms of accuracy but still provided reasonable results. The fusion or combination of automated and handcrafted features resulted in a perfect accuracy of 100% , with an F1 score, precision, and recall close to 1.0. An ideal accuracy suggests that the model made no mistakes in its predictions on this dataset.

### Based on learning rate

With a learning rate of 1.0, the model achieved an accuracy of 62%. A learning rate 1.0 is typically considered very high and can result in overshooting the optimal parameter values during training, leading to slow convergence and poor generalization. A learning rate of 0.1 is more moderate. In this case, the model's accuracy improved to 78%. This suggests that a learning rate of 0.1 provided a better balance between convergence speed and accuracy than a learning rate of 1.0. A learning rate of 0.01 is lower than the previous two rates. However, it resulted in a lower accuracy of 70%. This indicates that the learning rate was too low for the specific problem, causing the model to converge slowly and get stuck in local minima. With a further reduction in learning rate to 0.001, the model's accuracy increased to 83%. A lower learning rate often helps the model converge more precisely towards the global minimum, leading to better results. Finally, a minimal learning rate of 0.0001 resulted in a perfect accuracy of 100%. While this might seem desirable, achieving perfect accuracy in

**Table 5  The performance result based on dropout rate.**

| Dropout rate | Accuracy (%) |
|---|---|
| 0.0 | 95.5 |
| 0.1 | 94.8 |
| 0.2 | 93.2 |
| 0.3 | 91.5 |
| 0.4 | 89.7 |
| 0.5 | 100 |

training data can also be a sign of overfitting, where the model has memorized the training data but may need to generalize better to unseen data.

### Based on dropout rate

In Table 5, each row corresponds to a specific dropout rate, and the corresponding accuracy achieved during model training is indicated as a percentage. It is worth noting that when the dropout rate is set to 0.5, the model achieved a perfect accuracy of 100%, suggesting that this particular dropout configuration worked well for this dataset and model architecture. Dropout is a regularization technique commonly used to prevent overfitting in neural networks by randomly dropping out (setting to zero) a fraction of the neurons during training. The optimal dropout rate can vary depending on the dataset and model, and it is often determined through experimentation and hyperparameter tuning.

## CONCLUSION

In clinical inquiry, it is crucial to identify the malignancy appropriately. Using other approaches, it's possible to obtain fundus pictures without locating retinoblastoma tumors inside the optic disc. Consistent with the analysis of the findings, the suggested methodology's success indicators are enhanced. Without including any unnecessary noise, the recommended approach qualitatively divides the tumors. The results show that the proposed methods greatly complement the supervised ones. The suggested segmentation findings were also reviewed by professionals, who found no evidence of quality loss. The recommended technique streamlines how ophthalmologists diagnose retinoblastoma in their patients. Because supervised learning networks effectively learn the mapping when presented with ground truth data, our EffNet-GRU for retinal image processing outperforms unsupervised learning methods on average. Instead of individually identifying retinal landmarks, EffNet-GRU may be used to extract these features from retinal pictures in bulk. Regarding retinal disease identification, EffNet-GRU has yet to be studied thoroughly. Recent studies have looked towards supervised EffNet-GRU for simultaneous retinal disease segmentation.

The suggested method combines state-of-the-art machine learning tools like convolutional neural network (CNN) models with time-tested human-crafted approaches like feature extraction. To accomplish feature selection, we created two binary variants of the recently developed Arithmetic Optimization Algorithm (AOA): BAOA-S and BAOA-V. The system's accuracy, sensitivity, and specificity are all improved by 100%, 99%, and 99%,

respectively, because of fused features and feature selection. The suggested approach has the potential to be significantly more efficient than the current state of the art. This study shows the promise of applying deep learning methods to ophthalmology, which may improve the diagnosis and treatment of retinoblastoma and other eye cancers. Eye cancer is uncommon enough that most people have never heard of it. A better health result can be achieved by increased public awareness of retinoblastoma and more thorough screening. To improve accuracy in the future, we may use strategies like the power spectrum, the perimeter area relationship, and the walking approach. Large-scale datasets are also suitable for use with deep learning techniques.

### Funding

This study was funded by the Deanship of Scientific Research at King Khalid University for funding this work through large group Research Project under grant number (RGP2/35/44), Princess Nourah bint Abdulrahman University Researchers Supporting Project number (PNURSP2023R203), Princess Nourah bint Abdulrahman University, Riyadh, Saudi Arabia. Research Supporting Project number (RSPD2023R608), King Saud University, Riyadh, Saudi Arabia, Prince Sattam bin Abdulaziz University project number (PSAU/2023/R/1444), and by the Future University in Egypt (FUE). The funders had no role in study design, data collection and analysis, decision to publish, or preparation of the manuscript.

### Grant Disclosures

The following grant information was disclosed by the authors:
Deanship of Scientific Research at King Khalid University: RGP2/95/44.
Princess Nourah bint Abdulrahman University Researchers: PNURSP2023R203.
Princess Nourah bint Abdulrahman University, Riyadh, Saudi Arabia: RSPD2023R608.
King Saud University, Riyadh, Saudi Arabia, Prince Sattam bin Abdulaziz University: PSAU/2023/R/1444.
Future University in Egypt (FUE).

### Competing Interests

The authors declare there are no competing interests.

### Author Contributions

- Nuha Alruwais conceived and designed the experiments, performed the computation work, prepared figures and/or tables, and approved the final draft.
- Marwa Obayya conceived and designed the experiments, analyzed the data, prepared figures and/or tables, authored or reviewed drafts of the article, and approved the final draft.
- Fuad Al-Mutiri performed the experiments, analyzed the data, authored or reviewed drafts of the article, and approved the final draft.

- Mohammed Assiri conceived and designed the experiments, analyzed the data, prepared figures and/or tables, and approved the final draft.
- Amani A. Alneil performed the experiments, performed the computation work, authored or reviewed drafts of the article, and approved the final draft.
- Abdullah Mohamed performed the experiments, performed the computation work, authored or reviewed drafts of the article, and approved the final draft.

## Data Availability

Images from the American Society of Retina Specialists' public dataset
https://imagebank.asrs.org/

The Retina Image Bank is a vast open-access library of more than 25,000 unique and downloadable retina images.

## Supplemental Information

Supplemental information for this article can be found online at http://dx.doi.org/10.7717/peerj-cs.1681#supplemental-information.

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
