# Peer review of "Advancing retinoblastoma detection based on binary arithmetic optimization and integrated features"

_PeerJ Computer Science, doi:10.7717/peerj-cs.1681_

## Round 0.1 · original submission · Major Revisions

The reviewers have substantial concerns about this manuscript. The authors should provide point-to-point responses to address all the concerns and provide a revised manuscript with the revised parts being tracked in a different color.

**Language Note:** The review process has identified that the English language must be improved. PeerJ can provide language editing services - please contact us at copyediting@peerj.com for pricing (be sure to provide your manuscript number and title). Alternatively, you should make your own arrangements to improve the language quality and provide details in your response letter. – PeerJ Staff

·

Basic reporting

(1)The English used in this paper needs significant improvement. I suggest the manuscript be proof-read by an expert in the field of deep learning to check grammar and improve readability.
(2)The Introduction does not show context well. The objective and specific contribution of this paper are not clearly described.
(3)The raw data is not clearly introduced. For example, what’s the number of the original cases/images before augmentation? How are the CT and MRI data used in the proposed method?

Experimental design

(1)The major problem with experiments is the lack of comparison to baseline methods. As a paper proposing a new algorithm/classification-framework, it is essential to show that the proposed method is superior to existing methods.
(2)Ablation study is also needed to show the importance of key components of the proposed framework and the rationality of major design choice.
(3)Experiments on more datasets are needed and a dataset is usually divided into train/validation/test sets instead of only train-test sets.

Validity of the findings

The proposed method lacks of technical novelty because it is only an integration of several existing widely-used methods. The experimental design has some flaws, such as the lack of a validation set, and the finding is not very sound and robust.

Additional comments

This paper proposes a framework to detect and classify retinoblastoma. From a technical point of view, the proposed framework is a simple combination of existing algorithms and lacks of novelty. The experimental results are not sufficient to support the conclusion because of the flaws in experimental design, the lack of comparison to existing methods and the limited datasets used. The English has big room of improvement.

Reviewer 2 ·

Basic reporting

Placement of Data Augmentation in Pre-processing: Regarding the organization of the paper's content, I propose relocating the discussion of data augmentation techniques to the pre-processing section. This adjustment would align the paper's structure with common practices in presenting the sequence of data manipulation steps. By addressing data augmentation alongside other pre-processing methodologies, readers can gain a holistic understanding of the entire preparatory pipeline before model implementation.

Experimental design

1. Ablation Experiment for Feature Fusion Clarification: To further reinforce the significance of the fusion between hand-crafted and deep-learning features, I recommend the authors consider conducting an ablation experiment. By systematically removing one feature type at a time and assessing the impact on performance metrics, the study can provide a more robust justification for the integration of these feature categories. This analysis would enrich the discussion around the interplay between hand-crafted and deep-learning features, contributing to a deeper understanding of their individual and combined contributions to the model's success.

2. Elaboration on Dataset Splitting and Training Procedure: It would be valuable for readers to gain a comprehensive understanding of the paper's experimental setup. Could the authors elaborate on the methodology employed for splitting the dataset into training, validation, and test subsets? Additionally, delving further into the training process—such as the optimization algorithm used, learning rate schedule, and convergence criteria—would provide a clearer picture of how the model was fine-tuned. Providing this level of detail would enhance the reproducibility and rigor of the study.

Validity of the findings

3. Thorough Analysis of Results, Comparisons, and Overfitting Concerns: The reported 100% accuracy is indeed remarkable, prompting the need for a more in-depth discussion surrounding this achievement. To contextualize the findings, I suggest the authors undertake a thorough comparative analysis with similar research in the field. Comparisons with state-of-the-art models or benchmarks would not only demonstrate the paper's contributions but also shed light on its performance relative to existing solutions. Furthermore, addressing potential overfitting concerns and detailing how the model's generalization capabilities were ensured during training would augment the validity of the outcomes.

Additional comments

Open-sourcing of Dataset and Code: In the spirit of promoting transparency and facilitating further research in the domain, I encourage the authors to consider open-sourcing both their dataset and code. By making these resources publicly available, the wider research community can verify and build upon the work, fostering collaboration and accelerating advancements in the field. This step would contribute to the paper's impact and assist in replicating the experiments.

---

## Round 0.2 · accepted · Accept

Reviewers are happy with the revisions and I concur to recommend accepting the current version.

·

Basic reporting

Compared to the original manuscript, the revised version has made significant improvements in English. The author has made significant improvements to the introduction section which clearly demonstrates the objective and specific contribution of this paper. In the revised manuscript, the author has supplemented the the number of the original cases/images and the CT and MRI data used in the proposed method.

Experimental design

In the revised manuscript, the author provided reasonable explanations and improvements to the experimental design.

Validity of the findings

Conclusions are well stated, linked to original research question & limited to supporting results.

Additional comments

In summary, the author has fully responded to the review comments and made significant modifications to the original manuscript

Reviewer 2 ·

Basic reporting

The revised manuscript used clear and professional English. And it satisfies the publish criteria and answers my question.

Experimental design

The revised manuscript has a good experimental design.

Validity of the findings

The revised manuscript has a good validity of the findings.

Additional comments

I agree to accept this manuscript.